# Unveiling the Role of Contingent Negative Variation (CNV) in Migraine: A Review of Electrophysiological Studies in Adults and Children

**DOI:** 10.3390/biomedicines11113030

**Published:** 2023-11-11

**Authors:** María E. de Lahoz, Paloma Barjola, Irene Peláez, David Ferrera, Roberto Fernandes-Magalhaes, Francisco Mercado

**Affiliations:** Department of Psychology, School of Health Sciences, Universidad Rey Juan Carlos, 28922 Madrid, Spain; mariaeugenia.delahoz@urjc.es (M.E.d.L.); paloma.barjola@urjc.es (P.B.); irene.pelaez@urjc.es (I.P.); david.ferrera@urjc.es (D.F.); roberto.fernandes@urjc.es (R.F.-M.)

**Keywords:** Contingent Negative Variation (CNV), migraine without aura, migraine cycle, children, cortical hyperexcitability, S1–S2 paradigm, cognitive anticipation, chronic pain

## Abstract

Migraine has been considered a chronic neuronal-based pain disorder characterized by the presence of cortical hyperexcitability. The Contingent Negative Variation (CNV) is the most explored electrophysiological index in migraine. However, the findings show inconsistencies regarding its functional significance. To address this, we conducted a review in both adults and children with migraine without aura to gain a deeper understanding of it and to derive clinical implications. The literature search was conducted in the PubMed, SCOPUS and PsycINFO databases until September 2022m and 34 articles were retrieved and considered relevant for further analysis. The main results in adults showed higher CNV amplitudes (with no habituation) in migraine patients. Electrophysiological abnormalities, particularly focused on the early CNV subcomponent (eCNV), were especially prominent a few days before the onset of a migraine attack, normalizing during and after the attack. We also explored various modulatory factors, including pharmacological treatments—CNV amplitude was lower after the intake of drugs targeting neural hyperexcitability—and other factors such as psychological, hormonal or genetic/familial influences on CNV. Although similar patterns were found in children, the evidence is particularly scarce and less consistent, likely due to the brain’s maturation process during childhood. As the first review exploring the relationship between CNV and migraine, this study supports the role of the CNV as a potential neural marker for migraine pathophysiology and the prediction of pain attacks. The importance of further exploring the relationship between this neurophysiological index and childhood migraine is critical for identifying potential therapeutic targets for managing migraine symptoms during its development.

## 1. Introduction

Migraine is a common neurological disorder characterized by recurrent attacks of intense and throbbing headache, which can last for several hours or days [1]. In addition to pain, the migraine is often accompanied by other symptoms such as nausea, vomiting and hypersensitivity to sensory stimuli (photo, osmo or phonophobia) [1,2]. Patients also report other disabling disturbances, such as physical, cognitive and emotional alterations, either before, during or after headache attacks [3]. This concomitant symptomatology presents itself in a cyclical manner through a series of phases. Thus, the migraine cycle begins with a preictal or prodromal phase, involving physical and emotional symptoms experienced a few days or hours before the pain onset. It is followed by an ictal phase, when the throbbing pain occurs. Finally, the cycle ends with the resolution of the pain during the postdromal phase [3]. Particularly in migraine with aura, there occurs the so-called aura, a distinguishable phase in which transient neurological, visual, somatosensory, motor and speech symptoms usually appear [3]. In addition, a variable period of time (or interictal phase) without the presence of clinical symptomatology occurs between attacks [3] (a detailed graphical representation of the migraine cycle is displayed in Figure 1).

According to the third International Classification of Headache Disorders (ICDH-3), two main types of migraine can be diagnosed: migraine without aura and migraine with aura. The latter is characterized by the presence of the aura phenomenon. On the other hand, patients diagnosed with migraine without aura often refer to throbbing and unilateral pain and the presence of sensory hypersensitivity. Despite migraine—particularly migraine without aura [4,5]—constituting a common clinical condition, reaching high rates of prevalence in both children and adults (7–10% and 14%, respectively) [4], its pathogenesis is not yet fully understood, and its root cause remains unknown [2,6,7]. 

For several decades, both neural and vascular processes have been proposed to underlie the neurobiological substrate of migraine [2], based on of the Neuro-Vascular Theory [8]. This proposal assigned a critical role to the Trigeminal-Vascular System in the origin and maintenance of migraines, as well as many of their pain-related clinical symptoms, such as the throbbing nature of pain or extracranial allodynia, among others [9]. However, recent perspectives argue that migraine can be better understood as a ‘pure’ neuronal disorder [2,10]. The presence of cortical hyperexcitability along with the alteration of functional connectivity (desynchronization) among different brain regions [10,11,12] has led experts to consider migraine as an “altered brain state” [2,10,13]. It is thought that this altered state of the brain may be the result of functional and homeostatic changes in the brainstem and hypothalamus [12,13,14]. In particular, homeostatic imbalances in the synthesis and release of catecholamines at the brainstem level would lead to the hyperactivation of the dopaminergic and noradrenergic pathways as part of the trigeminal-thalamic-cortical loop [15,16]. These alterations have been observed in the functioning of the brain regions involved in the regulation of sensory information and pain signaling systems [12,13,14], which might contribute to the onset and persistence of the sensory hypersensitivity symptoms that characterize the migraine [11,12]. In this regard, several investigations have documented that migraine patients show an abnormal response to sensory stimulation, even during interictal periods [10,11,13,17]. 

Electroencephalographical (EEG) recordings have been the most frequently used methods to study the cortical excitability in migraine [12,17,18,19]. Specifically, event-related potentials (ERPs) have been examined, highlighting that patients exhibit abnormal neural responses that are suggestive of an impairment in the processing of information [11,13,20]. One of the most studied ERP components in migraine has been the Contingent Negative Variation (CNV) [20,21,22,23,24,25,26]. This waveform typically occurs due to the appearance of an expected stimulus (S2), signaled by a previous one (S1) (see Figure 2: CNV during the S1–S2 expectancy paradigm). CNV is characterized by a slow and late negative shift composed of two different subcomponents or phases of processing: (1) the early CNV (eCNV) that occurs between 550–750 ms from the S1 onset; (2) the late CNV (lCNV), a more prolonged wave beginning around 800 ms after the S1 onset [27,28]. The eCNV (maximal at frontal scalp areas) is enhanced by tasks or events that require the allocation of anticipatory attention, such as those involving emotional features, as is the case for negative or pain-related stimulation [29,30,31,32,33]. The data from migraine investigations have related the eCNV to the level of cortical excitability underlying the activation of the striato-thalamo-cortical loop [19,34,35]. On the other hand, lCNV (maximal at central scalp sites) has been related to motor preparation to upcoming stimulation. Nevertheless, the functional meaning of CNV in migraine is far from being defined. 

Several investigations have indicated that patients with migraine, particularly adults suffering from migraine without aura, exhibit two well defined phenomena: (1) higher CNV amplitudes [20,21,24,25,36,37] and/or (2) a deficit of CNV habituation [23,38,39,40,41] compared to healthy individuals. Hence, both CNV signals have been proposed as potential neural markers associated with upcoming migraine attacks [36,39,40]. However, to date, some inconsistencies are still observed with respect to the modulation of CNV at several levels. For instance, the CNV subcomponents (eCNV and lCNV) seem to be differently affected in migraine, and its relationship with the clinical symptomatology of the disease is still unclear. In addition, it has been observed that variations in the CNV amplitude and habituation are not constant throughout the migraine phases and the progression of the disease [36,38,39,40], showing higher amplitudes and reduced habituation during the days prior to the onset of an attack [36,39,40] and re-establishing in the ictal and postdromal phases [40], which has not been given a functional interpretation. 

Furthermore, it is important to consider the potential modulating effect of different factors, such as hormonal imbalance, psychological influences, pharmacological treatments and even genetic vulnerability aspects, on the extent of CNV in migraine and its clinical manifestations [23,24,42,43,44,45,46,47,48,49,50,51,52,53,54,55]. Endogenous variations in hormone levels (i.e., menstrual cycle and pregnancy) appear to influence brain electrophysiological activity and have an impact on the CNV amplitudes and habituation in women suffering from migraine as a function of their oestrogen levels [46,56,57,58]. Concerning the role of psychological factors, stressful, uncertain or threatening situations have also been associated with higher CNV amplitudes in migraine sufferers [24,44,57,59]; however, inconclusive data regarding their relationship with CNV, as well as their clinical implications in the pathology, have hindered the delimitation of solid conclusions.

On the other hand, the use of preventive medication treatments has been considered as a potential modulator factor of CNV in migraine. In this vein, drugs involved in neuronal excitability processes, such as anticonvulsants and beta-blockers [42,60,61], or those acting on serotonin regulation (e.g., triptans) [62,63], have been linked to modulation of CNV subcomponents for migraine patients. However, the effects of different pharmacological treatments on CNV are mixed. While beta-blockers induce overall CNV amplitude improvements [42,60], anticonvulsants and triptans act specifically on the eCNV [61,63] or lCNV [63]. In contrast, the phenomenon of CNV dishabituation seems to be effectively modulated by anticonvulsants drugs [61], while the use of beta-blockers and triptans to reduce it has been unsuccessful compared to placebo or other type of treatment [42,60,62,63]. Finally, it has been suggested that genetic vulnerability is likely to be an important factor in migraine, as reported by some familial studies. In this regard, CNV amplitudes have been found to be similar between healthy children and their parents with migraine [47]. Moreover, healthy siblings also exhibit comparable CNV amplitudes (particularly, eCNV) to their siblings with migraine [48,64]. Contrary to this findings, other investigations conducted with asymptomatic first-degree relatives of migraine patients showed undistinguishable eCNV amplitudes and habituation patterns [65]. 

Migraine is a disorder that can onset in early childhood and can be diagnosed as early as 5 to 6 years of age [4,49]. The few studies conducted in this population show similar findings related to the increased amplitudes and/or loss of habituation of CNV [47,48,50,51,66]. However, the results seem less conclusive, if possible [51,52], and this may due to the ongoing brain developmental processes during childhood [50,51,52,66]. Longitudinal studies have found that children with migraine exhibit atypical and even reversed maturation of CNV components at early ages compared to control children [51,52,66]. The great variability in the results could mean that the CNV is modulated by the developmental stage and is highly age dependent. Nevertheless, the available evidence in children with migraine makes it difficult to draw definitive conclusions about the functional role of CNV [50,51,66]. 

The current scientific evidence highlights the relationship between CNV and migraine without aura. However, given the varying and contradictory findings regarding the sensitivity of the CNV to sensory stimulation in migraine, it is important to clarify its contribution in the pathology and the extent of the potential modulatory factors that may account for these divergent results. Therefore, a review seems necessary and justified in this still under-explored field of research. Hence, the present review attempts to gain a more comprehensive and precise understanding of the functional meaning of CNV (both components: eCNV and lCNV) in migraine during the migraine cycle, considering different age groups (adults and children), and to further explore the role of potential modulatory factors. To the best of our knowledge, these issues have not previously been explored in migraine patients.

## 2. Materials and Method

This study was conducted in accordance with the Preferred Reporting Items for Reviews and Meta-analysis (PRISMA) guidelines [53]. This review was not previously registered.

### 2.1. Selection Criteria 

Primary studies were selected when the following criteria were satisfied: (1) experimental studies included, at least, a group of patients suffering from migraine without aura; (2) the studies used an expectancy paradigm (S1–S2) for the acquisition of the CNV component; (3) CNV amplitudes were reported; (4) data from both adult and children patients were considered to compare CNV amplitudes; (5) only findings reported in the English language were considered. Studies using non-English language or including reviews and single case reports were excluded. Articles with no full text availability were also excluded. No restrictions on publication date, sociodemographic factors (sex, ethnicity, age) or diagnostic classification were applied (e.g., ICDH-II, ICDH-III).

### 2.2. Information Sources and Search Strategy

The exhaustive scientific literature search to explore the relationship between CNV and migraine was conducted until September 2022. The search was performed in three databases (PubMed, SCOPUS and PsycINFO) using a combination of previously identified search terms: (1) “Migraine Disorders” and “Headache”’ (2) “Contingent Negative Variation”; (3) “Electroencephalography”. The Boolean operators “AND” and “OR” were used. 

Two sequential searches were performed in each database to mitigate the publication bias, ensure more exhaustive results and enhance the scientific rigor. The first search involved a free text search, where the search terms were entered without the use of Boolean operators. The second search was conducted using a more systematic approach, employing controlled vocabulary MeSH terms (PubMed and SCOPUS) and Thesaurus APA (PsycINFO), along with the appropriate Boolean operators, as follows: 

PubMed and SCOPUS: (((“Migraine Disorder”[Mesh]) OR “Headache”[Mesh]) AND “Contingent Negative Variation”[Mesh]) AND “Electroencephalography”[Mesh]))). 

PsycINFO: ((MM “Migraine Headache”) OR (DE “Headache”)) AND (MM “Contingent Negative Variation”) AND (DE “Electroencephalography”). 

This dual search strategy was conducted independently by two researchers (M.E.D.L.H and P.B.V.). In addition, a new search was carried out in the months prior to the writing of the manuscript (until April 2023) to explore the possible recent publication of eligible articles, but none were found. 

### 2.3. Studies Inclusion Procedure and Data Extraction

First, the title and abstract of the selected articles were screened after the search on each database to determine the eligibility criteria. In cases where the eligibility criteria could not be determined based only on the title or the abstract, full texts were also examined. Duplicates were removed in intra- and inter-searches for each database. Once the duplicates had been removed, the full-text articles were screened. Extracted data included the year of publication, author first name, characteristics of the sample (patients and control), the study design and the main electrophysiological results related to CNV. The reference list of the studies included in the current review was checked for the exploration of any additional study not previously found in the database searches. 

## 3. Results

The systematic literature search strategy yielded a total of 1111 scientific articles, of which 837 were excluded after the title and abstract screening, and 199 were duplicates (intra and inter search database). Therefore, 75 articles were further checked for their eligibility and inclusion in the review. However, 17 of them were excluded because the full-text was not available (key authors were contacted for providing the full-text, but no responses were received). After a deeper review of their content, another 24 articles were also removed due their failure to meet the inclusion criteria (see Table 1). Finally, 34 full-text articles were retrieved and considered relevant for further analysis and data extraction (Table 2). The flow diagram of the systematic search procedure is displayed in Figure 3.

Among the 34 selected articles, 24 (70.58%) included adult samples, 6 of them (17.64%) used child samples and 4 (11.76%) included mixed samples (adults and children). Concerning those articles that analyzed CNV-related data in adult patients with migraine without aura, 20 included a control group composed of healthy individuals, and 9 of them included an additional subgroup with another type of migraine (migraine with aura or chronic migraine) or tension headache. Only 4 studies used a single sample composed of patients with migraine without aura. Regarding the child samples studies, 5 of them included a healthy control group and 3 of them used an additional subgroup with a different type of migraine (migraine with aura or chronic migraine) or tension headache. Only 1 study used a single sample of children with migraine without aura. Regarding the patient gender, 29 articles included a mixed sample (both female and male or boys and girls, in the case of children’s samples) and 5 studies had only a group of women with migraine. It is also important to mention that 2 studies reported longitudinal results, while the rest were cross-sectional investigations.

According to the aim of the current review, the findings relevant to achieving a more precise and comprehensive understanding of the characteristics and functional role of CNV in migraine, as well as its potential moderators, will be presented in the following subsections. Due to the special characteristics of CNV in children, these results will be shown separately from those of adults with migraine. In addition, a graph summarizing the results on amplitude and habituation can be found for the adult and child samples (Figure 4 and Figure 5).

### 3.1. CNV Results in Adult Migraine Patients

The set of articles analyzed in adult patients (24 studies) provided data on the CNV amplitude, but only 18 of them also reported data on CNV habituation towards auditory (tones) or visual (light flashes) sensory stimulation. Overall, the obtained results clearly support that CNV amplitudes for migraine patients were higher (i.e., more negative) compared to control participants, especially in migraine without aura [21,23,24,36,37,38,39,40,44,45,56,57,58,59,60,61,65,88,89]. Nevertheless, some investigations reported inconsistent findings. The first studies conducted in this field reported higher CNV amplitudes for migraine, but the analyses did not distinguish between CNV subcomponents [20,21]. Although some further studies reported CNV differences between migraine patients and controls for both subcomponents (eCNV and lCNV), most of these investigations (15 articles) described that the enhancement of CNV amplitude was specially detected on the early subcomponent [23,24,36,38,39,40,44,45,56,57,58,61,65,88,89]. On the other hand, only a smaller group of studies described the effects on the lCNV [36,39,59,88]. Finally, the other investigations did not reveal enhanced amplitudes in any of the CNV subcomponents [20,26,62,68]. Indeed, the most recent study in this field, conducted by Tian and colleagues (2019) [26], found no significant differences in the CNV components (eCNV, lCNV or total CNV) between patients with migraine and healthy individuals [26].

Complementarily, 13 out of the 18 studies that analyzed CNV habituation reached significant statistical differences, showing a deficit in habituation for patients with migraine. This lack of habituation was only confirmed for the eCNV subcomponent [21,23,36,37,38,39,40,44,57,58,61,65,89]. That is, whereas the eCNV amplitudes did not show any change after the repetition of the stereotyped stimulation in migraine patients, the control participants exhibited a progressive decrease in eCNV amplitudes as the number of experimental trials increased [37,38,52]. This finding has been linked to a potential deficit to habituate towards sensory events. By contrast, some investigations conducted in this field failed to observe differences in the CNV habituation between healthy people and migraine [26,45], or such differences arose in association with the intake of some types of medication (it will be explained in more detail later) [42,60,63].

Based on the current findings, both CNV signals (increased eCNV amplitudes and the lack of eCNV habituation) have been proposed as potential neural markers associated with upcoming migraine attacks [39,40], the duration of the pathology and its chronification [38,39]. In this vein, it has been demonstrated that the migraine duration correlates with distinct abnormalities of eCNV. It has been observed that when a migraine has a chronic character (more than 15 attacks per month) or a prolonged duration (>120 months), the loss of habituation in the eCNV to sensory stimulation is even more prominent [38,39]. However, the eCNV amplitudes results have shown mixed evidence regarding the duration of the pathology. Thus, whereas Siniatchkin and colleagues (1998) [39] revealed that the chronicity of migraine symptoms led to only slightly higher eCNV amplitudes in these patients [39], further studies, such as the one conducted by Kropp and colleagues (2015) [38], confirmed that long-duration patients with migraine showed more pronounced overall CNV amplitudes [38]. 

The capability of various factors to modulate CNV amplitude/habituation, such as those related to the cyclic fluctuations of migraine, hormonal imbalance, psychological variables, pharmacological treatments or genetic vulnerability, was also reviewed. Thus, 20 of the 24 selected articles focused on such questions in adult patients [23,24,36,40,42,44,45,47,48,56,57,58,59,60,61,62,63,64,65,89]. Considering the cyclic fluctuation of migraine, three studies observed that the CNV underwent fluctuating changes relating to the migraine phases (i.e., interictal, preictal, ictal) [23,36,40], but the two subcomponents of CNV were not uniformly affected [23,36,40]. In particular, it was observed that the eCNV reached its most negative amplitude a few days before the onset of a migraine attack [23,40], and it normalized during and after the attack (a decrement of eCNV amplitude), showing amplitude values comparable to healthy individuals [23]. A similar pattern has been described for the habituation of the eCNV. Thus, the loss of habituation was clearly observed a few days before the ictal period (peaking the day before), returning to a normal habituation process during and after the migraine attack [23,40]. However, a more recent study conducted by Tian and colleagues (2019) [26] raised some doubts about the existence of CNV changes associated with the cyclic pattern of migraine [26]. They found no differences in any of the sub-components of CNV as a function of the number of days prior to a migraine attack when compared to healthy people. Furthermore, they referenced that enhanced amplitudes of eCNV along with a loss of habituation may result from several complex interactions between the intrinsic cerebral, hormonal and external environmental elements that act on genetically susceptible nervous systems [26], as other studies have shown [47,48,64,65]. Along this line, the transmission of CNV characteristics in migraine families may be genetically determined [48,65]. The analysis of families in which migraine is presented demonstrated that there are close similarities in the morphology and habituation of the early CNV component between children and parents with migraine [48]. Moreover, it was observed that asymptomatic first-degree relatives of patients with migraine exhibited higher eCNV amplitudes and a comparable level of habituation to patients with migraine [48,65]. Interestingly, the eCNV amplitudes were positively correlated with the number of family members suffering from migraine (i.e., the greater the number of affected individuals in the family, the more pronounced the eCNV abnormalities were detected in asymptomatic relatives) [65]. 

Regarding the hormonal influences, the data are somewhat conflicting. Hormonal changes seem to modulate the CNV amplitudes in female patients. In particular, it has been observed that women with migraine may exhibit higher amplitudes of the eCNV according to their oestrogen levels [56,57]. However, such findings are inconsistent due to higher amplitudes being reported during both low [56] and high [57] oestrogen levels. In addition, during pregnancy, no significant changes in the eCNV amplitudes were found in patients with migraine compared to healthy women, despite the hormonal fluctuations inherent to gestation. By contrast, differences in the habituation phenomenon have indeed been clearly identified during the pregnancy period, regardless of hormonal fluctuations [58]. Pregnant women with migraine normalized the CNV habituation pattern together with a decrease in clinical migraine symptoms (i.e., fewer pain attacks), but this favorable situation was reversed after the delivery, with the recurrence of migraine symptoms and CNV abnormalities [58]. 

Psychological factors (i.e., stress studies) have also been linked to CNV in migraine, although only four of the studies reviewed focused on it. Experimentally induced stress (e.g., tasks requiring rapid responses), a typical precipitant of migraine attacks, and the use of non-adaptive cognitive strategies, such as rumination, led to a more pronounced neurophysiological reactivity of the CNV. Sniatchkin and colleagues (2006) [24] described that subjects with migraine showed a greater amplitude of the eCNV component and a greater reduction in its habituation under stressful conditions (when they had to give faster responses) compared to control participants. In a further study, changes in CNV habituation in women with migraine could not be replicated [57]. When analyzing the use of coping strategies, another group of researchers observed that migraine patients tended to use ineffective cognitive coping strategies in the face of stressful situations. In particular, migraine patients who scored high in the use of cognitive rumination presented higher amplitudes of eCNV [44]. Finally, another study in which the uncertainty context was manipulated (i.e., the warning signal -S1- may or may not provide information about the imperative stimulus -S2-) showed that both the informative and non-informative warning elicited the same response in migraine patients; that is, the informative signal did not elicit higher amplitudes of the eCNV as it did in the healthy group [59]. 

Furthermore, the effects of pharmacological methods to prevent and relieve the pain symptoms associated with migraine attacks were also explored in relation to the CNV wave. A total of five studies [42,60,61,62,63] examined the effects of treatments based on pharmacological interventions on CNV amplitudes and habituation. The use of preventive medications, such as beta-blockers (e.g., Propranolol, Metoprolol) [42,60] and anticonvulsants (e.g., Topiramate, Levetiracetam) [61], was shown to be highly effective in the management of migraine symptoms, as well as in the modulation of CNV. Specifically, two studies using beta-blockers demonstrated efficacy in the reduction in the overall CNV amplitudes (but not on CNV habituation) in patients with migraine compared to pre-treatment and placebo conditions [42,60]. Notably, one of them reported that patients showing higher CNV amplitudes before treatment tend to respond better to beta-blockers than patients exhibiting lower CNV amplitudes at the pre-treatment phase, resulting in a noticeable restoration of the CNV amplitudes and symptoms in this group of patients [60]. Only one study using anticonvulsants showed post-treatment decrements in CNV amplitudes, mainly observed on the eCNV component, along with a lower frequency of migraine attacks [61]. Moreover, it also was observed that anticonvulsants improved eCNV habituation after treatment [61]. Nevertheless, the use of triptans (serotonin agonists) for acute migraine showed mixed benefits with respect to the CNV and migraine symptoms [62,63]. The only investigation showing the effects of triptans in migraine patients did so on both the eCNV and lCNV subcomponents [63]. However, other studies detected no differences in the CNV amplitude/habituation after triptan treatment compared to the placebo condition in women with migraine [62,63].

Finally, three studies that used non-pharmacological pain relief interventions exhibited promising results in both the clinical symptomatology and the restoration of the CNV signal. Clinical practices, such as progressive muscle relaxation [45], meditation [44] or aerobic exercise [89], have demonstrated positive effects, leading to a decrease not only in eCNV amplitudes, but also in the total amount of days with pain and the frequency of migraine attacks [44,45,89]. These types of strategies also showed a significant improvement in the CNV habituation [44,89], with the exception of progressive muscle relaxation [45].

### 3.2. CNV results in Child-Adolescents Migraine Patients

As can be observed in Table 2, only six studies explored the relationship between CNV and childhood migraine. All of them explored the CNV amplitudes [25,41,43,51,52,66], with five focusing on the study of CNV habituation to auditory (tones) or visual (light flashes) stimulation [25,41,43,52,66]. Overall, the studies using children migraine samples reported enhanced CNV amplitudes (especially in migraine without aura) compared to controls [25,41,43,51,52,66]. Similarly to adult patients, this effect was mainly detected for the early subcomponent of the CNV [25,41,43,66]. However, two investigations reported differences for both the eCNV and lCNV components [51,52]. On the other hand, several studies revealed the presence of a potential deficit in CNV habituation to sensory events in children with migraine without aura compared to controls or migraine with aura children. This lack of habituation has only been confirmed for the eCNV subcomponent [25,41,43,52]. Only one investigation failed to find differences in the habituation of the CNV between patients and healthy controls [66]. 

Similarly, to the adult studies, it has been demonstrated that migraine duration correlates with the distinct abnormalities observed in CNV. In this vein, children with a worse evolution of the pathology (duration of symptoms or frequency of attacks) showed more negative amplitudes and a more pronounced loss of habituation of the CNV compared to children showing an improvement or remission of migraine attacks and healthy children [25]. Furthermore, the effects of the cyclic fluctuations of migraine on the CNV amplitude have been also explored in children. Siniatchkin and colleagues (2000) [41] observed changes in the eCNV associated with the cyclical phases of migraine. In particular, children showed the highest CNV amplitudes, along with a loss of habituation, on the day before the onset of the attack, following by a normalization of the eCNV during the attack itself and the day after it [41]. Such a decrease in eCNV amplitude occurred abruptly, coinciding with the values recorded in healthy children, in contrast to those shown by adults, where abnormal eCNV amplitudes reversed progressively [41].

Some investigations have proposed the influence of psychosocial and genetic vulnerability factors as possible modulators of CNV in children with migraine. One study exploring CNV changes within migraine-affected families reported that both amplitude and habituation abnormalities of the CNV were equivalent between children with migraine and their parents with migraine [47]. In addition, this study showed that enhanced CNV amplitudes were also present in siblings without migraine [47]. A further investigation did not find any effect of genetic vulnerability factors on the eCNV amplitudes [64]. On the other hand, parents tend to exert more control over the behavior of their children with migraine compared to parents with healthy children. This increased control and the directive and specific interactions given by parents to their children with migraine correlated with the presence of a greater habituation loss and higher CNV amplitudes in these children [64]. Furthermore, behavioral training of exposure to aversive stimuli in children with migraine seems to improve children’s ability to cope with stressful situations and influences CNV abnormalities. Significant decreases in CNV amplitudes have been reported along with clinical improvements as a consequence of this type of training [43]. 

## 4. Discussion

The main question we aimed to explore in this review was focused on the comprehensive investigation of CNV features for a detailed understanding of its functional significance in migraine. We covered the differences in amplitude and habituation of the CNV (early and late subcomponents: eCNV and lCNV) observed in both adults and children with migraine, while examining the role of different factors (those related to cyclic phases of migraine, genetic vulnerability, hormonal imbalance, behavioral influences and pharmacological treatments) as potential modulators of this electrophysiological index. Most of the existing scientific reports repeatedly showed the presence of increased amplitudes along with an habituation deficit in the CNV for patients with migraine compared to the healthy population [21,23,24,36,37,38,39,40,44,45,56,57,58,59,60,61,65,88,89]. Although several studies of migraine suggest a relationship between changes in the CNV and cortical excitability alterations [17,19,34], none of them provided clear evidence or an explanatory theory concerning the functional involvement of the CNV in migraine [63]. In the following, we will try to provide an integrative and reasoned explanation for the cognitive function of the CNV in migraine. 

The current findings are quite consistent across studies with respect to the presence of CNV amplitude abnormalities in migraine that are more clearly detected in the eCNV subcomponent [23,24,36,38,39,40,44,45,56,57,58,61,65,88,89], being not as solid for lCNV [45,58]. Accordingly, habituation deficits of CNV have only been found for the eCNV subcomponent [21,23,36,37,38,39,40,44,57,58,61,65,89]. Interestingly, both the highest amplitude and the lowest values of the habituation of the eCNV have been associated with the fluctuating changes related to the migraine phases (i.e., interictal, preictal, ictal) [23,36,40]. In particular, the eCNV reached its greatest values a few days before the onset of a migraine attack [23,40]. Convergent data have been also reported with respect to the loss of CNV habituation, being more prominent a few days before the ictal period and peaking the day before. The different functional meanings attributed to each CNV subcomponent (eCNV and lCNV) [27,90,91,92] might account for the present data, where the CNV subcomponents are differently modulated in migraine. As mentioned above, the initial phase of the CNV (eCNV) has been considered as a neural correlate of cortical excitability and anticipatory attention that is modulated by properties conveyed by the warning stimulus in the S1–S2 paradigms (its amplitude seems to increase when individuals anticipate the appearance of significant stimulation, such as emotional [27,28,29,31,33,92,93] or threat/pain-related [93,94,95,96,97]). In this vein, the use of cues indicating the upcoming appearance of painful stimulation (i.e., electrical or ischemic induced pain) has been linked to higher amplitudes of overall CNV and eCNV than non-painful cued stimulation [70,97,98,99]. Although the evidence is still scarce, the higher amplitudes detected on the eCNV to upcoming stimulation support the role of this subcomponent in anticipatory attention for chronic pain patients [100,101]. Predicting future nociceptive stimuli seems to involve the activation of attention mechanisms that play an important role in the enhancement of pain perception [70,97,98]. Considering the previous findings, both higher eCNV amplitudes and the deficit of eCNV habituation in migraine patients could serve as specific and sensitive predictive indices of the proximity and periodicity of new migraine attacks [23,39,40]. Thus, the proximity of new attacks might lead to the greater allocation of attentional resources toward painful information in migraine patients [34,56,57]. On the other hand, the normalization of the eCNV during and after migraine attacks could represent a decrease in attention away from pain, as a brain protecting mechanism from noxious influences and overstimulation [102]. However, the scarce research focusing on the emotional and attentional aspects impacting CNV modulation presents a barrier to achieving a comprehensive understanding of the pain anticipatory mechanisms in migraines, despite the great importance of these processes in pain perception. Investigating the attentional and emotional processes in migraine could shed light on the potential critical influences on the eCNV and derive clinical implications on the evolution and expression of migraine. 

The data on CNV and its possible functional significance in migraine deserve further reflection. As mentioned above, the CNV has been linked to cortical excitability in migraine. Indeed, it has been argued that a complex interaction of neural mechanisms underlies the altered cortical excitability in migraine, including CNV abnormalities, along with other neurobiological changes [77,103]. Under this umbrella, the catecholaminergic pathway [21,56,63,77,102] appears to play a key role in the transmission and modulation of pain perception in specific brain regions of migraine patients, such as the striato-thalamo-cortical system [16,27,104,105]. Thus, the pattern observed in the catecholamine release across the different phases of the migraine cycle could be related to the changes detected in the CNV deflections, supporting the functional meaning of the CNV as a neurophysiological marker of migraine attacks and symptomatology [34,77,103,104]. Consistent with this point of view, the enhanced eCNV amplitudes and habituation deficits observed during the interictal and/or preictal period would be correlated with an increase in catecholaminergic (noradrenergic) [56,63,77,102] and a decrease in serotonergic activity [34,103,106], leading to increased cortical excitability [63,102]. Immediately after the onset of a migraine attack, the catecholamine levels are reversed by an increase in serotonergic transmission and, consequently, a decrement of cortical excitability is detected, along with similar eCNV values to those of healthy people [106,107]. Along this line of argument, the most common pharmacological agents in migraine prevention, such as anticonvulsants, antidepressants, beta-blockers and calcium channel blockers, have been repeatedly used for their impact on cortical excitability [42,60,61,62,63]. These pharmacological treatments acting on the catecholinergic and/or serotonergic signaling pathways have been shown to reverse cortical excitability, along with positive effects on patients’ clinical symptoms, including reductions in the CNV or eCNV waves (amplitude and habituation) [42,60,61,62,63]. In particular, beta-blockers were linked to changes in cortical excitability, normalizing cortical information processing and decreasing the vulnerability of the brain to migraine precipitants [42,108]. Overall, these findings suggest the presence of a homeostatic imbalance in the catecholaminergic and serotonergic pathways of migraine patients underlying a state of cortical hyperexcitability that could be reflected in those changes described in the CNV.

According to the results of the present review and the data reported by other studies, hormonal fluctuations in women with migraine, specifically the variation in oestrogen levels, markedly influence cortical excitability and the further occurrence of migraine attacks. Evidence from neurophysiological studies has confirmed that declining oestrogen levels modulate the activity of several neurotransmitter systems and the functioning of the pain-related neural networks implicated in the pathophysiology of migraine [109]. Thus, as the oestrogen levels decrease, a reduction in the functioning of the serotonin receptors and synthesis is detected [46,109,110]. These neurobiological changes have been associated with an enhancement of CNV amplitudes and a deficit in its habituation, increasing the risk of experiencing migraine episodes [56,57]. Electrophysiological data obtained from oscillatory based analyses studies have observed changes in the alpha and beta power in the frontal and parietal areas, which are also associated with a decrease in oestrogen levels in female migraine patients [46,110,111]. These findings suggest that only when oestrogen levels fall does cortical hyperexcitability and an increased risk of migraine episodes arise [46,112]. This neurobiological pattern has also been reported in pregnant women with migraine following delivery [58,113,114]. 

Additionally, these biological mechanisms could also be at play through the presence of behavioral triggers. It is known that one of the most common triggers for migraine is stress [115]. Stressful and uncertain situations, including inappropriate coping strategies, may increase cortical excitability (through the activity of noradrenergic system) and contribute to the electrophysiological changes observed in migraine [24,44,57,59,116,117,118,119,120]. In addition to stressful situations, other psychological conditions might modulate the electrophysiological activity associated with CNV in migraine. In this context, migraine is often associated with psychiatric disorders, such as anxiety and depression. People affected by migraine often exhibit heightened susceptibility to anxiety and depression, which, in turn, can contribute to an increased risk of migraine attacks, perpetuating a cycle of heightened anxiety and depressive symptoms [121,122,123,124,125]. This comorbidity may share a common pathogenic mechanism involving various brain regions and neurotransmitters pathways, where serotonin may be playing a pivotal role for CNV modulation [30,126].

Studies conducted in migraine families have reported similar electrophysiological altered patterns (higher eCNV amplitudes and loss of habituation) between family members with migraine and those who did not show clinical symptoms [48,65]. Several studies [47,48,65,77,78,127] have provided evidence of the influence of familial/genetic factors on the abnormal pattern of cortical excitability, at least in adults with migraine [50,52]. Perhaps the genetic factor involved in the alteration of cortical excitability could contribute, to some extent, to the variability in the CNV amplitude, although it is not yet known which genes are involved in this potential influence. This fact could be relevant for individuals who do not exhibit overt migraine symptoms, such as in the case of children who do not show evident symptoms but have a family history of the disease. The presence of this genetic vulnerability may or may not trigger the onset of migraines, depending on external factors [43,47,65]. Although further research is needed, the evidence suggests that psychosocial events and family factors are related, to some extent, to CNV abnormalities in migraine, but the degree of causality between them is not fully established.

Evidence from children with migraine confirms the presence of some CNV abnormalities already reported in adult patients (at least partially) [25,41,43,66], but these electrophysiological alterations are not yet well defined. It is important to consider some aspects related to the brain maturation development to better understand the CNV findings in children. Electrophysiological activity, and hence cortical excitability, seems to be age-dependent [50,51,52,66]. Usually, the CNV measured in children reflects a higher amplitude and weaker habituation compared to that obtained in adults [50,51,52,66], but these differences progressively begin to diminish until early adulthood, consistent with the natural maturation of the brain [48,50,128]. Nevertheless, this developmental pattern associated with CNV appears to be distinctive in children with migraine [50,51,52,66] where the increase in eCNV amplitudes occurs at younger ages, around 6 years old [66]. However, between 10 and 12 years old, there is an inversion in the eCNV pattern, showing a moderate decrease in its amplitude [51,52], which contrasts with children without pathology and adult patients with migraine [50]. This fact seem to be associated with additional nonspecific subcortical activation in the brainstem, along with an enhancement of catecholaminergic activity [66,129]. As suggested by previous investigations, the increase in the eCNV amplitudes at early ages could contribute to the predisposition and manifestation of migraine attacks in adulthood, given that these eCNV abnormalities are equally observed after 30 years old [50,51,52]. Although speculative and pending confirmation, this distinctive electrophysiological pattern that seems to occur in children with migraine could serve as a marker of migraine predisposition. However, the limited number of studies conducted in children, along with their small sample sizes, poses difficulties for extracting solid conclusions about the effects on the CNV (eCNV/lCNV) in children with migraine [50,51,52,66]. This scarcity is surprising, considering that the CNV is susceptible to anticipatory processes related to pain perception [51,52]. 

Although we have conducted a comprehensive review, it has not been without some limitations that could partly explain the variability of the obtained results. The studies reviewed using the S1–S2 paradigm to evoke the CNV have shown a great diversity of sensory stimulation, different time intervals between cue and target stimuli and multiple scalp locations where electroencephalography activity was recorded, among other methodological features. The use of both auditory and visual stimuli (flashes) varied depending on the study analyzed, as did their presentation times, which ranged between 25 ms and 200 ms. Likewise, another difference between the studies has been observed in the inter-stimulus interval (ISI) between S1–S2 stimuli, where the CNV component is detected. In this regard, the ISIs ranged from 1–4 s, and this could have influenced the reliability of the CNV characterization. Differences in how the component was recorded and analyzed also showed variations that could affect the characterization of this neural index. Most of the studies recorded the CNV and its subcomponents on the Cz electrode. This localization does not cover the topographic characteristics of the CNV and its subcomponents, as it is well-defined that the CNV has a fronto-parietal distribution (more frontal the eCNV and more central-parietal the lCNV) [29,130,131]. Finally, the temporal window used to analyze this component of the ERPs (overall CNV) has been defined differently across studies (e.g., 0–3000 ms; 800–1000 ms; 1800–2000 ms; 500–3000). In particular, the variability in the time windows chosen for the eCNV subcomponent ranged between 500 ms and 1500 ms (e.g., 550–750 ms; 500–1500 ms; 600–1100 ms; 700–1100 ms). This methodological variety may contribute to the inconsistencies of the reported results and the difficulty in establishing more solid conclusions concerning the role of the CNV in childhood migraine.

## 5. Conclusions

To the best of our knowledge, this is the first review of CNV findings in migraine. The current results strongly support the presence of electrophysiological abnormalities (an enhancement of eCNV amplitude along with a deficit in its habituation) in migraine patients compared to healthy individuals, which are influenced by both external and internal factors. These abnormalities are more prominent a few days before the onset of a migraine attack, potentially serving as specific and sensitive indices for predicting migraine attacks. However, in the case of children with migraine, the evidence is limited, making it challenging to interpret the results and understand the functional significance of the CNV during childhood. The differences between children and adults with migraine in CNV studies might be explained by the maturation of the developing brain.

In summary, further research on this topic, particularly in pediatric samples, is needed to gain a better understanding of the functional role and potential clinical implications of the CNV in migraine. This approach could lead to a more accurate diagnosis, prediction and even prevention of migraine attacks, as well as, ultimately, to the identification of potential therapeutic targets for effective strategies in the treatment of migraine symptoms. Furthermore, it underlines the importance of investigating other factors with possible modulatory influences on the CNV component and migraine pathology, such as the impact of emotional and anticipatory attention processes, among others.

## Figures and Tables

**Figure 1 biomedicines-11-03030-f001:**
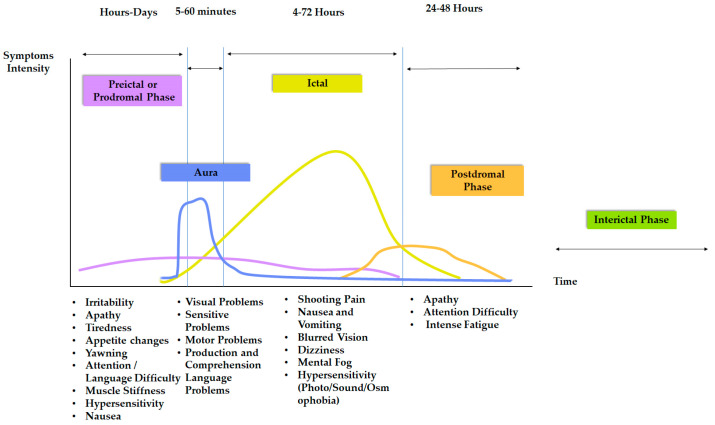
Representation of different phases of migraine cycle, along with the most common symptoms and duration.

**Figure 2 biomedicines-11-03030-f002:**
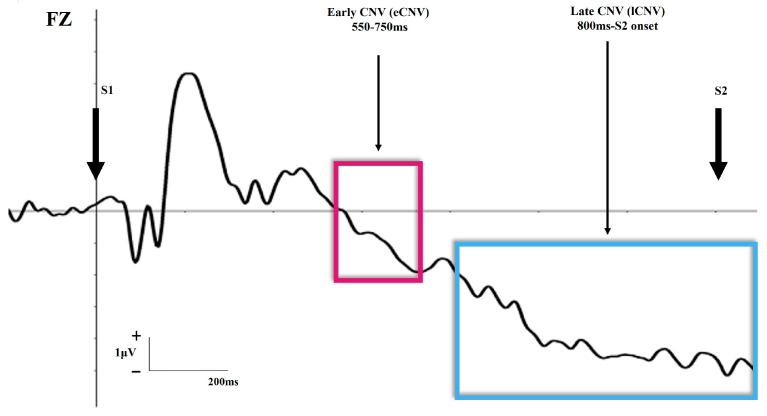
Graphical schematic representation (in Fz electrode) of the latency, polarity and amplitude of early CNV and late CNV ERP components during an S1–S2 task. The purple square represents the early CNV (550–750 ms) and the blue square represents the late CNV (800 ms-S2 onset).

**Figure 3 biomedicines-11-03030-f003:**
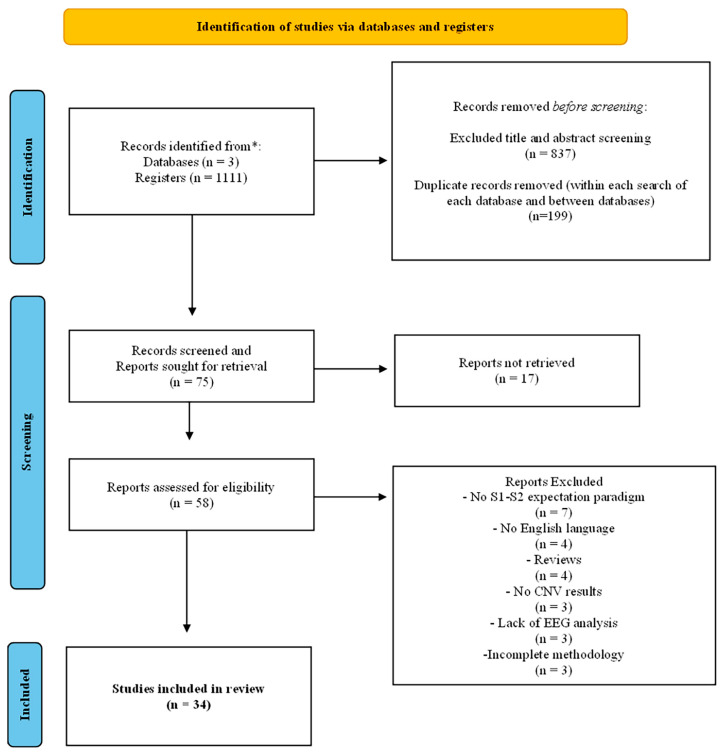
The PRISMA Flow Diagram for Literature Search in the Review. (* PubMed, SCOPUS and PsycINFO databases).

**Figure 4 biomedicines-11-03030-f004:**
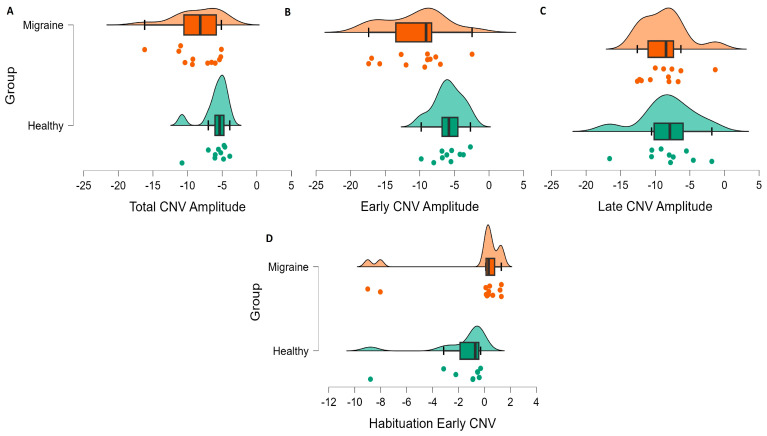
Summary graphs depicting CNV amplitude and habituation data from studies involving adult migraine patients and healthy controls. Each graph displays the mean of (**A**) Total CNV amplitudes, (**B**) Early CNV amplitudes, (**C**) Late CNV amplitudes and (**D**) Early CNV habituation for each reviewed study (points). In the center of each graphical representation, a box-and-whisker provides essential statistics, including the median (indicated by a thickened black bar), quartile distribution (displayed as a box) and data variance (illustrated by the whiskers) for each group (migraine patients and healthy controls). At the top of each graphical representation, a scatter curve depicts the data distribution for each group.

**Figure 5 biomedicines-11-03030-f005:**
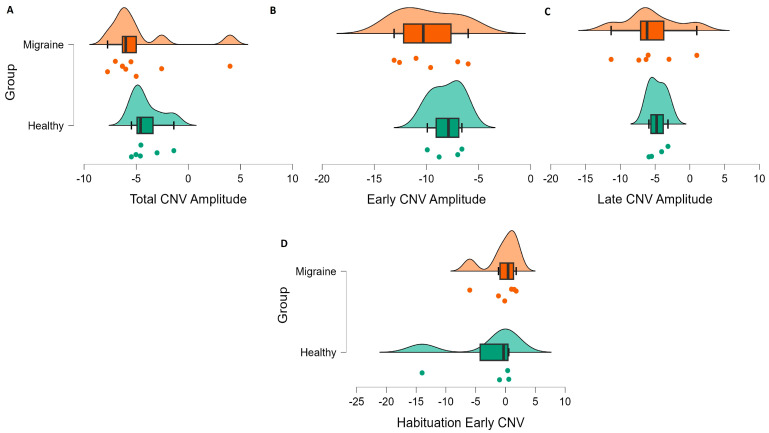
Summary graphs depicting CNV amplitude and habituation data from studies involving children with migraine and healthy controls. Each graph displays the mean of (**A**) Total CNV amplitudes, (**B**) Early CNV amplitudes, (**C**) Late CNV amplitudes and (**D**) Early CNV habituation for each reviewed study (points). In the center of each graphical representation, a box-and-whisker provides essential statistics, including the median (indicated by a thickened black bar), quartile distribution (displayed as a box) and data variance (illustrated by the whiskers) for each group (migraine patients and healthy controls). At the top of each graphical representation, a scatter curve depicts the data distribution for each group.

**Table 1 biomedicines-11-03030-t001:** Articles excluded for the Review and the reason of exclusion.

Articles	Reason of Exclusion
Schoenen (1986) [54] Dixon (1999) [55] Siniatchkin, et al. (2000) [67] Müller, et al. (2002) [68] Kropp, et al. (2002) [69] Babiloni, et al. (2004) [70] Ozkan, et al. (2012) [71]	Without S1–S2 expectation paradigm
Timsit, et al. (1987) [72] Gerber, et al. (1993) [73] Kropp, et al. (2005) [74] Meyer, et al. (2018) [75]	Articles were not in English language
Kropp, et al. (1993) [22] Schoenen, et al. (1993) [76] Gerber, et al. (1998) [77] Coppola, et al. (2012) [78]	Reviews
Smite, et al. (1994) [79] Bender, et al. (2006) [80] Lev, et al. (2013) [81]	No results related to CNV
Besken, et al. (1993) [82] Ahmed, (1999) [83] Harmela, et al. (2017) [84]	Incomplete statistical or EEG analysis
De Noordhouth, et al. (1987) [85] Kropp, et al. (2000) [86] Bender, et al. (2005) [87]	Inadequate study methodology

**Table 2 biomedicines-11-03030-t002:** Relevant data extracted from the selected articles.

Authors (Year)	Sample Characteristics	Use of Medication	Experimental Task	Electrode’s Location	CNV Temporal Window (ms)	Significant Differences in CNV (Yes/No)	CNV Amplitude Results		CNV Habituation Results
Maertens de Noordhout, et al. (1986) [21]	79 Migraine Group (MG)/Tension Headache (TH) 6 Classic Migraine/with Aura (WA) 23 Common Migraine/WoA 16 Combined, Mostly Migraine 21 Tension Headache 13 Combined, Mostly Tension Headache 33 Healthy Control (HC)	No prophylactic treatment Analgesic	48 Trials Warning Tones (S1) Target Flashes (S2)	Unspecified	Contingent Negative Variatiob (CNV) (Baseline-1000 ms)	Yes	Pure Migraine > Controls or Tension (***) Combined Mostly Migraine > Controls or Tension (***)		↓ habituation in Migraine
Schoenen, et.al, (1986) [60]	33 MG /WoA/ 27 Metoprolol 6 Propanolol	No prophylactic treatment	48 Trials Warning Tone (S1) Target Flashes (S2) 1 s Inter Stimulus Interval (ISI)	Unspecified	CNV (800–1000 ms)	Yes	MG after treatment < MG before treatment		No significant changes in CNV
Böker, et.al, (1990) [20]	17 MG 12 WoA 5 WA 8 HC	Unspecified	32 Trials Warning Tones (S1) Response Flashes (S2) 1 s ISI (CNV1) 3 s ISI (CNV3)	Cz, Fz, C3, C4	Early CNV (eCNV) (550–750 ms)	No	WoA > WA/HC (T)		*---*
					Late CNV (lCNV) (200 ms pre S2-2800–3000 ms)	No	WoA > WA/HC (T)		
Nagel-Leiby, et al. (1990) [56]	12 MG (Women) 7 WoA 5 WA 6 HC (Women)	No prophylactic and contraceptives treatment	48 Trials Warning Auditory (S1) Target Flashes (S2) 4 s ISI	Cz, Pz	eCNV (500–1500 ms)	Yes	WA > WoA Menses Phase		*---*
Göbel. et al. (1993) [62]	14 WoA (Women) Sumatriptan or Placebo	No prophylactic treatment	32 Trials Warning Auditory (S1) Target Flashes (S2) 2 s ISI	Cz	CNV (1800–2000 ms)	No	Sumatriptan = Placebo Interictal and Ictal Pre-Post Treatment		*---*
Kropp, et al. (1993) [37]	12 WoA 20 HC	Unspecified	40 Trials Warning Tone (S1) Imperative Tone (S2) 3 s ISI	Cz	CNV (0–3000 ms)	Yes	WoA > HC (**)		
					eCNV (550–750 ms)	No	WoA > HC (T)		↓ habituation eCNV WoA
					lCNV (200 ms pre S2-2800–3000 ms)	No	WoA = HC		
Kropp, et al. (1995) [36]	16 WoA 22 HC	No medication (Prophylactic or analgesic)	40 Trials Warning Tone (S1) Imperative Tone (S2) [-During Interictal -During Ictal]	Cz	CNV (0–3000 ms)	Yes	Interictal > Ictal (**) WoA Ictal < HC (**)		↓ habituation eCNV interictal WoA
					eCNV (550–750 ms)	Yes	WoA > HC (***) Interictal > Ictal (***) WoA Ictal = HC		
					lCNV (200 ms pre S2-2800–3000 ms)	Yes	Interictal > Ictal (**) WoA Ictal < HC (*)		
Kropp, et al. (1998) [23]	16 WoA 22 HC	No prophylactic treatment	40 Trials Warning Tone (S1) Imperative Tone (S2) 3 s ISI	Cz	eCNV (550–750 ms)	Yes	WoA > HC (−1) (***) WoA (−1) > WoA (+1) (***)		↓ habituation eCNV WoA Previous Day Ictal (−1) ↑ habituation eCNV WoA Following Day Ictal (+1)
Siniatchkin, et al. (1998) [39]	30 MG 15 WoA 15 Chronic Daily Headache (CDH) 15 HC	No prophylactic treatment	40 Trials Warning Auditory (S1) Imperative Auditory (S2) 3 s ISI	C3, C4	CNV (0–3000 ms)	Yes	WoA > CDH (**) WoA > HC (**)		↓ habituation eCNV WoA ↓ habituation eCNV CDH
					eCNV (550–750 ms)	Yes	WoA > CDH (***) WoA > HC (***)		
					lCNV (200 ms pre S2-2800–3000 ms)	Yes	CHD < WoA (**) CHD < HC (**)		
Kropp, et al. (1999) [50]	162 WoA 320 HC Age Subgroups 8–14 15–19 20–29 30–39 40–49 50–59	No prophylactic treatment	40 Trials Warning Tone (S1) Imperative Tone (S2) 3 s ISI	Cz	CNV (0–3000 ms)	Yes	WoA >HC (**)		
					eCNV (550–750 ms)	Yes	WoA > HC (***)		↓ habituation eCNV
					lCNV (200 ms pre S2-2800–3000 ms)	No	WoA = HC		
Kropp, et al. (1999) [47]	40 WoA: 14 Children WoA 26 Adult WoA 24 HC: 11 Children HC 13 Adult HC 5 Sibling Migraine Children (SMC)	No prophylactic treatment	40 Trials Warning Tone (S1) Target Tone (S2) 3 s ISI	Cz	CNV (0–3000 ms)	Yes	Children WoA = Adults WoA Children WoA/SMC > Children HC (*)		---
					eCNV (550–750 ms)	Yes	Children WoA = Adults WoA Children WoA > Children HC (*) Adult WoA > Adult HC (**)		
Siniatchkin, et al. (2000) [40]	20 WoA 12 HC	No prophylactic and contraceptives treatment	40 Trials Warning Auditory (S1) Imperative Auditory (S2) 3 s ISI	C3, C4	CNV (500–3000 ms)	Yes	WoA > HC (−1) (**)		↓ habituation eCNV WoA (−1)
					eCNV (550–750 ms)	Yes	WoA > HC (−1) (**)		
Siniatchkin, et al. (2000) [41]	10 Children WoA 20 Children HC	No prophylactic treatment Analgesics	Warning Auditory (S1) Imperative Auditory (S2) 3 s ISI	Cz	eCNV (550–750 ms)	Yes	Children WoA > Children HC (−1/Ictal/+1) (Maximum amplitudes −1)		↓ habituation eCNV Children WoA (Most pronounced deficit 1–2 days before attack)
Siniatchkin, et al. (2000) [48]	43 Families with Migraine: 45 Children WoA 36 Sibling Migraine Children (SMC) 30 Parents WoA 54 Healthy Parents 41 Healthy Families: 48 Children 82 Parents	No prophylactic treatment	40 Trials Warning Auditory (S1) Imperative Auditory (S2) 3 s ISI	Cz	eCNV (550–750 ms)	Yes	Children WoA > Healthy Parents Migraine Families (***) Children WoA > Healthy Parents Healthy Families (**) Greater values Children WoA Parents WoA = Healthy Parents		↓ habituation eCNV WoA Children (Migraine Children > Healthy Children > Migraine Adults)
					lCNV (200 ms pre S2-2800–3000 ms)	No	---		
Siniatchkin, et al. (2001) [65]	35 WoA: 35 Healthy Young Positive WoA Family 35 Healthy Young Negative WoA Family	No prophylactic treatment	40 Trials Warning Auditory (S1) Imperative Auditory (S2) 3 s ISI	Cz	eCNV (550–750 ms)	Yes	WoA > Negative WoA Family (***) Positive WoA Family > Negative WoA Family (***)		↓ habituation eCNV Positive WoA family ↓ habituation eCNV WoA Positive WoA > Negative WoA Family
Mulder, et al. (2001) [63]	20 WoA: Pre- Post Attack Sumatriptan 20 HC	Antidepressants B-Blockers Lithium	52 Trials Auditory Warning (S1) Flashes Response (S2) 3 s ISI	Fz, Cz, Pz	eCNV (550–750 ms)	Yes	WoA = HC WoA Post Attack Sumatriptan < HC (**) WoA Post Attack Sumatriptan < WoA Habitual Medication (***)		Habituation WoA Post Attack Sumatriptan = Habituation WoA Habitual Medication = HC
					lCNV (200 ms pre S2-2800–3000 ms)	Yes	WoA = HC WoA Post Attack Sumatriptan < HC (**) (Most prominent at the Frontal area)		
Bender, et al. (2002) [51]	61 Children WoA 76 Children HC	No prophylactic treatment	20 Trials Auditory Warning (S1) Auditory Imperative (S2) 3 s ISI	Cz	CNV (0–3000 ms)	Yes		Children WoA > HC (**)	---
					eCNV (550–750 ms)	No		Children WoA = Children HC	
					lCNV (200 ms pre S2-2800–3000 ms)	Yes		Children WoA > Children HC (**)	
Gerber, et al. (2002) [64]	30 Migraine Families WoA 30 Migraine Children 30 Migraine Mothers 28 Siblings Migraine Children (SMC) 20 Healthy Families 20 Healthy Children 20 Healthy Mothers	No prophylactic or acute treatment	40 Trials Auditory Warning (S1) Auditory Imperative (S2) 3 s ISI	Cz	CNV (500–3000 ms)	No	Migraine Children > Healthy Children (T)		↓ habituation eCNV Migraine Children = Healthy Children
					eCNV (550–750 ms)	No	Migraine Children > Healthy Children (T) Migraine Childrens > Sibling Migraine Children(T)		
Mulder, et al. (2002) [59]	20 MG 14 WoA 6 WA 22 HC	Antidepressants, B-Blockers or Lithium	Visual Warning (S1) Visual Response (S2) 3 s ISI [Certain/Uncertainty conditions]	Fz, Cz, Pz	eCNV (600–1100 ms)	No	HC = WoA/WA		---
					lCNV (200 ms pre S2-2800–3000 ms)	Yes	WoA < HC (**)		
Siniatchkin, et al. (2006) [57]	17 WoA (Women) 15 HC (Women)	No prophylactic medication No oral contraceptives	40 Trials Auditory Warning (S1) Auditory Imperative (S2) 3 s ISI (Stressful Condition) + [Premenstrual and Ovulatory Phases]	Cz	eCNV (550–750 ms)	Yes	WoA > HC (**) WoA Premenstrual > WoA Ovulatory (**) WoA Premenstrual + Stress > HC Premenstrual + Stress (**) WoA Premenstrual + Stress > WoA Ovulatory +Stress (**)		---
Siniatchkin, et al. (2006) [24]	45 MG 30 WoA 15 WA Pre-Ictal group (1–3 days before) Post-Ictal group (1–3 days after) Interictal group 20 HC	No prophylactic medication	40 Trials Auditory Warning (S1) Auditory Imperative (S2) 3 s ISI (Stressful Condition)	Cz	eCNV (550–750 ms)	Yes	MG Pre-Ictal + Stress > HC (**)		↓ habituation eCNV WoA Pre-Ictal + Stress
Bender, et al. (2007) [66]	101 Children MG 69 WoA 32 WA 23 Children TH 81 Children HC Prepubertal 6–11 Years Postpubertal 12–18 Years	No prophylactic medication	60 Trials Auditory Warning (S1) Auditory Imperative (S2) 3 s ISI	Cz/FCz/FC1/FC2/C3/C4/C5/C6/CP3/CP4/CP6/CP5/P3/P4	eCNV (700–1100 ms)	Yes	↑ WoA PrePubertal over Cz/FCz/FC1/FC2 (**) HC > WoA (**)		---
Siniatchkin, et al. (2007) [42]	20 WoA 10 Metoprolol 10 Placebo	No prophylactic medication	40 Trials Auditory Warning (S1) Auditory Imperative (S2) 3 s ISI	Cz	CNV (500–3000 ms)	Yes	Metoprolol < Placebo (**)		↑ habituation eCNV Metoprolol
Darabaneanu, et al. (2008) [58]	26 WoA/Mg (Women) 14 Pregnant Migraine 12 Non-Pregnant Migraine 30 HC (Women) 15 Pregnant Healthy 16 Non-Pregnant Healthy	No prophylactic medication	40 Trials Auditory Warning (S1) Auditory Imperative (S2) 3 s ISI	Cz	eCNV (550–750 ms)	No Yes	Pregnant Migraine = Pregnant Healthy (Third period of pregnancy) Pregnant Migraine > Pregnant Healthy (**) (After delivery)		↑ habituation eCNV WoA Pregnant
De Tommaso, et al. (2008) [61]	45 WoA: 18 Topiramate 18 Levetiracetam 9 Placebo 24 HC	No prophylactic medication	48 Trials Auditory Warning (S1) Auditory Imperative (S2) 3 s ISI	Fz, Cz, Pz	eCNV (550–750 ms)	Yes	MG > HC (**) Topiramate/Levetiracepam < Placebo (**)		↓ habituation eCNV WoA ↑ habituation eCNV WoA Post Treatment
Oelkers-Ax, et al. (2008) [52]	46 Children-Adolescent MG 28 WoA 18 WA 57 Children-Adolescent HC	No prophylactic medication	60 Trials Auditory Warning (S1) Auditory Imperative (S2) 3 s ISI	64 leads	CNV (0–3000 ms)	Yes	WoA < HC (***)		↓ habituation eCNV MG
					eCNV (550–750 ms)	No	WoA < HC (T)		
					lCNV (200 ms pre S2-2800–3000 ms)	Yes	WoA < HC (***)		
Siniatchkin, et al. (2010) [25]	27 Children WoA 9 Migraine Remission 12 Migraine Improved 6 Migraine Worsened 23 Children HC	No prophylactic medication	40 Trials Warning Auditory (S1) Imperative Auditory (S2) 3 s ISI	Cz	eCNV (550–750 ms)	Yes	Migraine Worsened > Migraine Remission (**) Migraine Worsened > HC (**) Migraine Remission = Healthy Controls		↓ habituation eCNV Migraine Worsened > Improved ↓ habituation eCNV Migraine Worsened > Healthy Children
Siniatchkin, et al. (2011) [43]	26 Children-Adolescents WoA 13 Behavioural Programme MIPAS 13 Biofeedback (BF) Treatment Group	No medication (Prophylactic or analgesic)	Auditory Warning (S1) Auditory Imperative (S2) 3 s ISI	FC1/FC2/FC5/FC6/CP1/CP2/CP5/CP6/TP9/TP10	eCNV (550–750 ms)	Yes	MIPAS < BF in frontal areas (**)		↑ habituation eCNV WoA MIPAS after treatment in frontal and central areas
					lCNV (200 ms pre S2-2800–3000 ms)	Yes	MIPAS < BF in central areas (**)		
Kropp, et al. (2012) [88]	24 WoA 24 HC	Unspecified	24 Trials Auditory Warning (S1) Auditory Target (S2) 3 s ISI	Cz, C3, C4	CNV (0–3000 ms)	Yes	WoA > HC (*)		---
					eCNV (550–750 ms)	Yes	WoA > HC (**)		
					lCNV (200 ms pre S2-2800–3000 ms)	Yes	WoA < HC (**)		
Overath, et al. (2014) [89]	28 MG Aerobic Endurance Program 22 WoA 6 WA	No medication	40 Trials Warning Auditory (S1) Imperative Auditory (S2) 3 s ISI	Cz	eCNV (550–750 ms)	Yes	After aerobic program < before aerobic program (**)		↑ habituation eCNV WoA After Aerobic Program
Kropp, et al. (2015) [38]	32 WoA 17 Short Duration of Migraine Disease (<120 months) 15 Long Duration of Migraine Disease (≥120 months) 16 HC	Unspecified	40 Trials Warning Auditory (S1) Imperative Auditory (S2) 3 s ISI	Cz	CNV (0–3000 ms)	Yes	WoA > HC (*)		↓ habituation eCNV WoA
					eCNV (550–750)	Yes	WoA > HC > Long Duration Disease (***)		
Keller, et al. (2016) [44]	46 MG 35 WoA 11 WA 45 Migraine Meditation Group 46 HC	No prophylactic medication	40 Trials Warning Auditory (S1) Imperative Auditory (S2) 3 s ISI *Stress Coping: SVF-78	Cz	CNV (0–3000 ms)	Yes	MG > Migraine Meditation (***) Migraine Meditation < HC (***) MG > HC (**)		↓ habituation eCNV MG
					eCNV (550–750 ms)	Yes	MG > Migraine Meditation (***) Migraine Meditation < HC (**) MG > HC (**)		
Meyer, et al. (2016) [45]	35 MG /WA/WoA/ 16 Migraine Progressive Muscle Relaxation (PMR) Training 19 Migraine Waiting-List 46 HC 21 Healthy PMR Training 25 Healthy Waiting List	No prophylactic medication	Acoustic Warning (S1)—Imperative (S2)	Cz	CNV (0–3000 ms)	Yes	Pre-PMR training: MG > HC (***)		No significant differences
					eCNV (550–750 ms)	Yes	Pre-PMR training: MG > HC (**) Post-PMR training: < Pre-PMR (**)		
Tian, et al. (2019) [26]	34 WoA 31 HC	No prophylactic medication	40 Trials Auditory Warning (S1) Auditory Imperative (S2) 2 s ISI	Fz, Cz, C3, C4	CNV (0–3000 ms)	No	WoA = HC		No significant differences
					eCNV (550–750 ms)	No			
					lCNV (200 ms pre S2-1800–2000 ms)	No			

MG = Migraine Group, WoA = Migraine Without Aura, WA = Migraine With Aura, TH = Tensional Headache, CDH = Chronic Daily Headache, HC = Healthy Control, SMC = Sibling Migraine Children, CNV = Contingent Negative Variation, eCNV = Early CNV, lCNV = Late CNV, S1 = Warning Stimulus, S2 = Imperative Stimulus, ISI = Inter Stimulus Interval, −1 = Previous Day Ictal, +1 = Following Day Ictal, (*) = *p* = 0.05, (**) = *p* = 0.005, (***) = *p* = 0.0001, (T) = Tendency, MIPAS = Migraine Patient Seminar for Families.

## Data Availability

Not applicable.

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
