# Peer review of "Unveiling the Role of Contingent Negative Variation (CNV) in Migraine: A Review of Electrophysiological Studies in Adults and Children"

_biomedicines, 2023, doi:10.3390/biomedicines11113030_

Round 1

Reviewer 1 Report

Comments and Suggestions for Authors

The present review addresses the prospect of contingent negative variation (CNV) in migraine without Aura. The topic is timely, although the following aspects need to be fixed.

In lines 118-120 and the discussion the authors emphasized the important aspect of the influence of hormonal effects on CNV. Please discuss/include recent studies addressing hormonal changes in women with migraine and its relation to excitability (https://doi.org/10.3390/app13137443 ). Although this findings are related to changes in EEG spectral power, it would be interesting to comment on the changes in power in relation to cortical excitability and CNV.

-I suggest the use of meta-statistics to have a measure of the effect regarding amplitude and habituation across studies.

-The review would benefit tremendously if the authors make use of graphs to summarize the results on amplitude and habituation and the quantification of variability of parameters across studies.

-The sample size of articles dealing with migraine in children is very limited, which represents a strong limitation when reporting effects on this cohort.  Please comment on this.

Comments on the Quality of English Language

The English quality is fine. 

Author Response

Consulte el archivo adjunto.

Reviewer 2 Report

Comments and Suggestions for Authors

The main purpose of the systematic review of de Lahoz Naveiro et al. entitled:”Unveiling the Role of Contingent Negative Variation (CNV) in  Migraine: A Systematic Review of Electrophysiological Studies  in Adults and Children” was to explore the connection between CNV as neurophysiological index and migraine without aura. Although, there are inconsistencies about CNS significance, the authors highlighted a role of the CNV as a potential neuronal marker for migraine pathogenesis and even prediction of pain attacks. In addition, the power of several modulatory factors was appropriately revised. Overall, I found the topic of the submitted review interesting. Migraine is the second most disabling condition worldwide and is a highly burdensome condition for patient, families, and society. However, I found some minor issues that need to be addressed by the authors.

MINOR

1. Migraine is comorbid with several psychiatric disorders including depression and anxiety disorders. Very often, each condition increases the risk of the other one. This aspect was not discussed despite it could draw attention of a broader scientific community. Therefore, I suggest including this issue in the manuscript.

2. In my opinion, the Discussion section appears overly lengthy and could benefit from being more concise. Rather than reiterating information from the Results section, the authors should focus on offering their personal interpretations of the gathered findings.

Round 2

Reviewer 1 Report

Comments and Suggestions for Authors

Thanks a lot for addressing my concerns. 

Comments on the Quality of English Language

The writing is suitable.

Author Response

We thank again reviewer 1 for the recommendation on the English editing. We have made a thorough revision to improve the English throughout the manuscript. The changes made in the revised version of the manuscript are highlighted using red text. We certainly expect that this modified version of the manuscript meets the language standards of Biomedicine.